# Entropy and Semi-Entropies of LR Fuzzy Numbers’ Linear Function with Applications to Fuzzy Programming

**DOI:** 10.3390/e21070697

**Published:** 2019-07-16

**Authors:** Jian Zhou, Chuan Huang, Mingxuan Zhao, Hui Li

**Affiliations:** 1School of Management, Shanghai University, Shanghai 200444, China; 2School of Economics and Management, Tongji University, Shanghai 200082, China

**Keywords:** fuzzy programming, entropy, semi-entropy, linear function, LR fuzzy number

## Abstract

As a crucial concept of characterizing uncertainty, entropy has been widely used in fuzzy programming problems, while involving complicated calculations. To simplify the operations so as to broaden its applicable areas, this paper investigates the entropy within the framework of credibility theory and derives the formulas for calculating the entropy of regular LR fuzzy numbers by virtue of the inverse credibility distribution. By verifying the favorable property of this operator, a calculation formula of a linear function’s entropy is also proposed. Furthermore, considering the strength of semi-entropy in measuring one-side uncertainty, the lower and upper semi-entropies, as well as the corresponding formulas are suggested to handle return-oriented and cost-oriented problems, respectively. Finally, utilizing entropy and semi-entropies as risk measures, two types of entropy optimization models and their equivalent formulations derived from the proposed formulas are given according to different decision criteria, providing an effective modeling method for fuzzy programming from the perspective of entropy. The numerical examples demonstrate the high efficiency and good performance of the proposed methods in decision making.

## 1. Introduction

Fuzzy programming (FP), a technique incorporating the concept of fuzziness in programming models, has extensive applications in the field of operations research (i.e., project planning [1,2], manufacturing [3,4], investment problems [5], etc.) due to its advantage in handling problems with ambiguity and vagueness, which are commonly met in practice. In the existing literature, various types of FP problems with fuzzy parameters and/or fuzzy variables [6] have been studied. To characterize the relationships between the variables, numerous statistical measures are utilized (e.g., expectation, variance), among which fuzzy entropy has had attention paid to it and has been widely accepted as an important measurement for weighing information quantity and denoting the uncertainty.

The birth of fuzzy entropy can be traced back to Zadeh [7], who initialized the fundamental fuzzy entropy, the weighted Shannon entropy [8], in an endeavor to enlarge the domain of applicability of classical entropy, and the superior performance of measuring the degree of fuzziness has made it a research hotpot under fuzzy environments quickly. Since then, various types of fuzzy entropy were put forward subsequently. De Luca and Termini [9] suggested the notion of entropy and four requirements that a fuzzy entropy measure should comply with, giving a way to measure the “indefiniteness”. A class of entropies in finite fuzzy sets and their properties were presented by Trillas and Riera [10]. Kosko [11] developed a general entropy measure based on an intuitive ratio of distances. Pal and Pal [12,13] introduced three types of entropy along with their applications to highlight their applicability to various problems (see also [14,15,16,17]).

Indeed, the above-mentioned measures are available with their own objectives and merits. However, they were based on the possibility measure presented in Zadeh [18,19], which fails the intuitive and significant property of self-duality and describes the uncertainty arising from linguistic ambiguity rather than information deficiency [20]. Considering that, Li and Liu [21] introduced a novel definition of fuzzy entropy in view of the self-dual credibility measure raised by Liu and Liu [22] to depict the uncertainty induced by information deficiency. Inspired by this new concept, theoretical research [20,23,24], as well as real applications [25,26,27,28,29,30,31] on credibility-based entropy have increased rapidly.

In the applications of entropy within the framework of credibility theory, portfolio optimization has occupied more than half of them. For instance, Huang [25,26] formulated some credibility-based mean-entropy models by employing fuzzy entropy as a risk measure and compared them with other models. Li et al. [29] introduced a credibilistic minimum entropy estimation in the application of fuzzy portfolio selection. Zhou et al. [31] developed a time-consistent multi-period rolling portfolio optimization model based on the credibilistic entropy. Since entropy expresses both high and low extreme uncertainty, whereas only one side is disliked in decision making, it can be improved and applied in more areas besides portfolio selection. In this paper, the lower and upper semi-entropies are set forth to deal with the two types of decision-making problems, that is return-oriented problems that focus on the downside uncertainty that may decrease the return, and cost-oriented problems, which care more about the upside uncertainty that may increase the cost, respectively.

On the other hand, in terms of theoretical research, the definitions of various types of entropy have been highly concerned, whereas the research on calculations is far more than enough. In view of this, the inverse credibility distributions (ICD) of some specific arithmetical operations based on the credibility measure introduced by Zhou et al. [32] are utilized in the present work to calculate the entropy and semi-entropies of the well-known LR fuzzy numbers, a type of fuzzy number defined by Dubois and Prade [33]. Aggregating the results into the FP models with entropy or semi-entropies as risk measures, four mean-entropy or mean-semi-entropy optimization models are formulated. Owning to the favorable linearity of the ICD demonstrated by Liu [34] and Yao and Ke [35], the formulas for calculating the entropy and semi-entropies of a linear function are derived for transforming the proposed optimization models into the equivalent forms, which can be solved in a much easier way. To verify and compare the practical applications of the proposed formulas and models, two common decision-making problems, portfolio selection and R&D product investment, are illustrated.

In summary, this paper is dedicated to broadening the application areas of the credibility-based entropy and to provide a framework for solving FP problems from the perspective of entropy and semi-entropies, which are calculated by a simplified method. The reminder of this paper is as follows. Section 2 reviews several concepts related to LR fuzzy numbers with some examples. Section 3 introduces the definition of credibility-based entropy, whereafter the entropies of regular LR fuzzy numbers and their linear functions are proposed and some related theorems are proven. Subsequently, the corresponding parts of semi-entropy are described in Section 4, in which the definitions of lower and upper semi-entropies are given so as to quantify one side uncertainty. The entropy optimization models are set forth and exemplified in Section 5. Finally, some conclusions are summarized in Section 6.

## 2. Preliminaries

In the following, some necessary concepts and operational laws of fuzzy set theory, which lay the foundation for the following sections, are reviewed successively, including the LR fuzzy numbers, credibility function (CF), credibility distribution (CD), inverse credibility distribution (ICD), and expected value.

**Definition** **1**(Dubois and Prade [33])**.**
*The left (or right) shape function L (or R) of an LR fuzzy number is a decreasing function from $R^{+}\to \left[0,1\right]$ satisfying the conditions below:*
*(1)* L(0)=1;*(2)* L(x)<1,∀x>0;*(3)* L(x)>0,∀x<1;*(4)* L(1)=0[orL(x)>0,∀xandL(+∞)=0].

**Definition** **2**(Dubois and Prade [36])**.**
*A fuzzy number δ is said to be of the LR type if the membership function (MF) is:*
(1)$$\mu \left(x\right)=\left\{\begin{array}{cc} L\left(\frac{\sigma -x}{\alpha }\right), & if\;x\leq \sigma \\ R\left(\frac{x-\sigma }{\beta }\right), & if\;x>\sigma , \end{array}\right.$$
*where α,β>0 are the left spread and right spread, and the real number σ is called the mean value. Typically, δ is denoted as ${\left(\sigma ,\alpha ,\beta \right)}_{LR}$.*

For the ease of distinguishing the definitions from the examples, the LR fuzzy number δ in Definition 2 is illustrated with some frequently-used ones with different types of MFs and the same parameter setting (a,b,c) in Examples 1–4, in which a,b,c stand for the mean value, the left spread, and the right spread, respectively.

**Example** **1.***If L(x)=R(x)=max{1−x,0}, then δ is a triangular fuzzy number and satisfies $\delta ∼T{\left(a,b,c\right)}_{LR}$ with the MF (see Figure 1a):*
(2)$$\mu \left(x\right)=\left\{\begin{array}{cc} \frac{x-a+b}{b}, & if\;a-b\leq x<a\\ \frac{a+c-x}{c}, & if\;a\leq x<a+c\\ 0, & otherwise. \end{array}\right.$$

**Example** **2.***A parabolic fuzzy number with $L\left(x\right)=R\left(x\right)=\max\left\{1-x^{2},0\right\}$ is denoted as $\delta ∼P{\left(a,b,c\right)}_{LR}$ with the MF (see Figure 1b):*
(3)$$\mu \left(x\right)=\left\{\begin{array}{cc} 1-{\left(\frac{a-x}{b}\right)}^{2}, & if\;a-b\leq x<a\\ 1-{\left(\frac{x-a}{c}\right)}^{2}, & if\;a\leq x<a+c\\ 0, & otherwise. \end{array}\right.$$

**Example** **3.***A normal fuzzy number with $L\left(x\right)=R\left(x\right)=\max\{{\left(1-x\right)}^{2},0\}$ is denoted as $\delta ∼N{\left(a,b,c\right)}_{LR}$, and the MF μ is (see Figure 1c):*
(4)$$\mu \left(x\right)=\left\{\begin{array}{cc} {\left(1-\frac{a-x}{b}\right)}^{2}, & if\;a-b\leq x<a\\ {\left(1-\frac{x-a}{c}\right)}^{2}, & if\;a\leq x<a+c\\ 0, & otherwise. \end{array}\right.$$

**Example** **4.***Let $L\left(x\right)=\max\left\{1-x^{2},0\right\}$ and $R\left(x\right)=\max\{{\left(1-x\right)}^{2},0\}$. Then, a mixture fuzzy number $\delta ∼M{\left(a,b,c\right)}_{LR}$ has the MF (see Figure 1d):*
(5)$$\mu \left(x\right)=\left\{\begin{array}{cc} 1-{\left(\frac{a-x}{b}\right)}^{2}, & if\;a-b\leq x<a\\ {\left(1-\frac{x-a}{c}\right)}^{2}, & if\;a\leq x<a+c\\ 0, & otherwise. \end{array}\right.$$

Assume that α is a real number and δ is a fuzzy number with the MF μ. With the consideration of self-duality, Liu and Liu [22] initialized that the fuzzy event {δ≤α} has the credibility of:(6)$$\mathrm{Cr}\left\{\delta \leq \alpha \right\}=\frac{1}{2}\left(\underset{x\leq \alpha }{\sup}\mu \left(x\right)+1-\underset{x>\alpha }{\sup}\mu \left(x\right)\right).$$

Accordingly, the CD and CF of a fuzzy number were defined as follows.

**Definition** **3**(Liu [22], Li [37])**.**
*Let δ be a fuzzy number. The CD Φ:R→[0,1] and the CF v:R→[0,1] of δ are defined, respectively, as:*
(7)Φ(x)=Cr{δ≤x},v(x)=Cr{δ=x},

In other words, Φ(x) refers to the credibility that the value of the fuzzy number δ is equal to or less than *x*, and v(x) is the credibility that the value of δ is equal to *x*.

**Remark** **1.***Regarding an LR fuzzy number $\delta ∼{\left(\sigma ,\alpha ,\beta \right)}_{LR}$ with the MF μ in Equation (Equation 1), its CD can be worked out in view of Equations (Equation 6) and (Equation 7) as:*
(8)$$\Phi \left(x\right)=\left\{\begin{array}{cc} \frac{1}{2}L\left(\frac{\sigma -x}{\alpha }\right), & if\;x\leq \sigma \\ 1-\frac{1}{2}R\left(\frac{x-\sigma }{\beta }\right), & if\;x>\sigma . \end{array}\right.$$
*and the CF v of δ is:*
(9)$$v\left(x\right)=\left\{\begin{array}{cc} \frac{1}{2}L\left(\frac{\sigma -x}{\alpha }\right), & if\;x\leq \sigma \\ \frac{1}{2}R\left(\frac{x-\sigma }{\beta }\right), & if\;x>\sigma . \end{array}\right.$$

It can be seen from Equation (Equation 8) that the CD of δ is a nondecreasing function, and Zhou et al. [32] proved that there are some favorable properties if it is a continuous and strictly increasing function. To study this special case, some new concepts concerning regular LR fuzzy numbers, as well as some related operational laws were raised by Zhou et al. [32].

**Definition** **4**(Zhou et al. [32])**.**
*It is said that an LR fuzzy number δ is regular if its CD *Φ* is regular, i.e., *Φ* is continuous and strictly increasing on {x|0<Φ(x)<1}, and the inverse function (IF) $\Phi^{-1}$ is called the ICD of δ.*

Zhou et al. [32] further verified that the LR fuzzy number $\delta ∼{\left(\sigma ,\alpha ,\beta \right)}_{LR}$ is regular when the shape functions are continuous and strictly decreasing, and its ICD is derived from Equation (Equation 8) as:(10)$$\Phi^{-1}\left(\gamma \right)=\left\{\begin{array}{cc} \sigma -\alpha L^{-1}\left(2\gamma \right), & \mathrm{if}\;\;\gamma \leq 0.5\\ \sigma +\beta R^{-1}\left(2-2\gamma \right), & \mathrm{if}\;\;\gamma >0.5. \end{array}\right.$$

Apparently, according to Definition 4, the four types of fuzzy numbers in the examples above (Examples 1–4) are regular.

Next, the operational law of calculating the ICD of a strictly monotone function with independent regular LR fuzzy numbers was provided.

**Theorem** **1**(Zhou et al. [32])**.**
*Let $\delta_{1}$, $\delta_{2}$, ⋯, $\delta_{n}$ be independent regular LR fuzzy numbers; their CDs are, respectively, $\Phi_{1}$, $\Phi_{2}$, ⋯, $\Phi_{n}$. If $f(x_{1},x_{2},\cdots ,x_{n})$ is a strictly increasing function with regard to $x_{1}$, $x_{2}$, ⋯, $x_{m}$ and a strictly decreasing function with regard to $x_{m+1}$, $x_{m+2}$, ⋯, $x_{n}$, then $\delta =f(\delta_{1},\delta_{2},\cdots ,\delta_{n})$ is a regular LR fuzzy number and has the ICD:*
(11)$$\Phi^{-1}\left(\gamma \right)=f\left(\Phi_{1}^{-1}\left(\gamma \right),\cdots ,\Phi_{m}^{-1}\left(\gamma \right),\Phi_{m+1}^{-1}\left(1-\gamma \right),\cdots ,\Phi_{n}^{-1}\left(1-\gamma \right)\right).$$

**Definition** **5**(Liu and Liu [22])**.**
*Let δ be a fuzzy number. Then, its expected value can be defined by:*
(12)$$E\left[\delta \right]=\;\;\int_{0}^{+\infty }\mathrm{Cr}\left\{\delta \geq t\right\}dt-\int_{-\infty }^{0}\mathrm{Cr}\left\{\delta \leq t\right\}dt,$$
*provided that at least one of the two integrals is finite.*

**Theorem** **2**(Liu and Liu [38])**.**
*Let $\delta_{1}$ and $\delta_{2}$ be independent fuzzy numbers with finite expected values. Then, for any real numbers $\lambda_{1}$ and $\lambda_{2}$, we have:*
(13)$$E\left[\lambda_{1}\delta_{1}+\lambda_{2}\delta_{2}\right]=\lambda_{1}E\left[\delta_{1}\right]+\lambda_{2}E\left[\delta_{2}\right].$$

**Theorem** **3**(Zhou et al. [32])**.**
*Let δ be a regular LR fuzzy number. If there exists the expected value of δ, then:*
(14)$$E\left[\delta \right]=\int_{0}^{1}\Phi^{-1}\left(\gamma \right)d\gamma$$
*where $\Phi^{-1}$ is the ICD of δ.*

## 3. The Entropy of a Linear Function with LR Fuzzy Numbers

In this section, the definition of entropy for fuzzy numbers is introduced, and then, the equivalent formulas of calculating the entropy of a regular LR fuzzy number and a linear function of regular LR fuzzy numbers are performed by virtue of the ICD. To verify the performance of the proposed formulas, some examples are illustrated.

### 3.1. The Definition of Entropy

Within the framework of credibility theory, Li and Liu [21] defined the entropy of a continuous fuzzy number as follows.

**Definition** **6**(Li and Liu [21])**.**
*Let δ be a continuous fuzzy number with a CF v. Then, the entropy of δ can be defined as:*
(15)$$H\left[\delta \right]=\;\;\int_{-\infty }^{+\infty }S\left(v(x)\right)dx,$$
*where the function S(t)=−tlnt−(1−t)ln(1−t), as shown in Figure 2.*

**Example** **5.***Given that $\delta ∼T{\left(a,b,c\right)}_{LR}$ is a triangular fuzzy number in Example 1 and Figure 1a, on the basis of Equations (Equation 9) and (Equation 15), the entropy of δ can be deduced as:*
(16)$$\begin{array}{cc} H[\delta ]\;\;\;\; & =\int_{a-b}^{a}S\left(\frac{1}{2}\times \left(1-\frac{a-x}{b}\right)\right)dx+\int_{a}^{a+c}S\left(\frac{1}{2}\times \left(1-\frac{x-a}{c}\right)\right)dx\\ & =\int_{0}^{0.5}2bS\left(t\right)dt-\int_{0.5}^{0}2cS\left(t\right)dt\\ & =\frac{b+c}{2}. \end{array}$$

**Example** **6.***Let $\delta ∼P{\left(a,b,c\right)}_{LR}$ be a parabolic fuzzy number in Example 2 and Figure 1b. It follows from Equations (Equation 9) and (Equation 15) that the entropy of δ is:*
$$\begin{array}{cc} H[\delta ]\;\;\;\; & =\int_{a-b}^{a}S\left(\frac{1}{2}-\frac{1}{2}{\left(\frac{a-x}{b}\right)}^{2}\right)dx+\int_{a}^{a+c}S\left(\frac{1}{2}-\frac{1}{2}{\left(\frac{x-a}{c}\right)}^{2}\right)dx\\ & =\int_{0}^{0.5}b{\left(1-2t\right)}^{-\frac{1}{2}}S\left(t\right)dt+\int_{0}^{0.5}c{\left(1-2t\right)}^{-\frac{1}{2}}S\left(t\right)dt\\ & =\frac{1}{6}\left(b+c\right)\left(8-\pi -2\ln2\right). \end{array}$$

### 3.2. The Entropy of a Regular LR Fuzzy Number

On the basis of the relationship between CF and CD, the entropy of a continuous fuzzy number can be calculated by means of its CD. In addition, for a regular LR fuzzy number whose ICD exists, the entropy can be figured out in a simple and convenient way. In this section, the calculating formulas are elaborated and proven as follows.

**Theorem** **4.***Given that δ is a continuous fuzzy number and its entropy exists, then the entropy is:*
(17)$$H\left[\delta \right]=\int_{-\infty }^{+\infty }S\left(\Phi \left(x\right)\right)dx,$$
*where *Φ* is the CD of δ.*

**Proof** **of Theorem 4.**Denote Φ(x) and v(x) as the CD and CF of a continuous fuzzy number ${\left(\sigma ,\alpha ,\beta \right)}_{LR}$, respectively. Based on Definition 3, it can be concluded that:
(18)$$\Phi \left(x\right)=\left\{\begin{array}{cc} v(x), & if\;x\leq \sigma \\ 1-v(x), & if\;x>\sigma . \end{array}\right.$$
By virtue of the equation S(t)=S(1−t) and Definition 6, we have:
$$\begin{array}{cc} \;H[\delta ]\;\;\;\; & =\int_{-\infty }^{+\infty }S\left(v(x)\right)dx=\int_{-\infty }^{\sigma }S\left(v(x)\right)dx+\int_{\sigma }^{+\infty }S\left(1-v(x)\right)dx=\int_{-\infty }^{+\infty }S\left(\Phi \left(x\right)\right)dx. \end{array}$$ □

**Theorem** **5.***Let δ be a regular LR fuzzy number. Then, the entropy of δ can be calculated as:*
(19)$$\;H\left[\delta \right]=\int_{0}^{1}\Phi^{-1}\left(\gamma \right)\ln\frac{\gamma }{1-\gamma }d\gamma ,$$
*where $\Phi^{-1}$ is the ICD of δ.*

**Proof** **of Theorem 5.**With a view toward Theorem 4, the entropy can be expressed as:
$$\begin{array}{cc} H[\delta ]\;\;\;\; & =\int_{-\infty }^{+\infty }S\left(\Phi (x)\right)dx=\int_{-\infty }^{0}\int_{0}^{\Phi (x)}S^{\prime }\left(\gamma \right)d\gamma dx+\int_{0}^{\infty }\int_{\Phi (x)}^{1}-S^{\prime }\left(\gamma \right)d\gamma dx, \end{array}$$
where $S^{\prime }\left(\gamma \right)={\left(-\gamma \ln\gamma -\left(1-\gamma \right)\ln\left(1-\gamma \right)\right)}^{\prime }=-\ln\frac{\gamma }{1-\gamma }$. By using the Fubini theorem, we have:
$$\begin{array}{cc} H[\delta ]\;\;\;\; & =\int_{0}^{\Phi (0)}\int_{\Phi^{-1}\left(\gamma \right)}^{0}S^{\prime }\left(\gamma \right)dxd\gamma +\int_{\Phi (0)}^{1}\int_{0}^{\Phi^{-1}\left(\gamma \right)}-S^{\prime }\left(\gamma \right)dxd\gamma \\ & =-\int_{0}^{1}\Phi^{-1}\left(\gamma \right)S^{\prime }\left(\gamma \right)d\gamma \\ & =\int_{0}^{1}\Phi^{-1}\left(\gamma \right)\ln\frac{\gamma }{1-\gamma }d\gamma . \end{array}$$ □

Compared with Definition 6 and Theorem 4, the formula put forward by Theorem 5 excludes the function S(t) and therefore has an advantage in reducing computational complexity, though it is only applicable to regular LR fuzzy numbers.

**Example** **7.***According to Equation (Equation 10), the ICD of a normal fuzzy number $\delta ∼N{\left(a,b,c\right)}_{LR}$ in Example 3 and Figure 1c can be derived as:*
$$\Phi^{-1}\left(\gamma \right)=\left\{\begin{array}{cc} a-b+b\sqrt{2\gamma }, & if\;\gamma \leq 0.5\\ a+c-c\sqrt{2-2\gamma }, & if\;\gamma >0.5. \end{array}\right.$$
*and then, it follows from Theorem 5 that the entropy of δ is:*
$$\begin{array}{cc} \;H[\delta ]\;\;\;\; & =\int_{0}^{0.5}\left(a-b+b\sqrt{2x}\right)\ln\frac{x}{1-x}dx+\int_{0.5}^{1}\left(a+c-c\sqrt{2-2x}\right)\ln\frac{x}{1-x}dx\\ & =\left(b+c\right)\int_{0.5}^{1}\ln xdx-\left(b+c\right)\int_{0}^{0.5}\ln xdx+\left(b+c\right)\int_{0}^{1}u^{2}\ln\left(\frac{u^{2}}{2-u^{2}}\right)du\\ & =\left(b+c\right)\left(\ln2+\frac{4}{3}-\frac{4\sqrt{2}}{3}\ln\left(1+\sqrt{2}\right)\right). \end{array}$$

**Example** **8.***With a view toward Equation (Equation 10), the ICD of a mixture fuzzy number $\delta ∼M{\left(a,b,c\right)}_{LR}$ in Example 4 and Figure 1d can be figured out as:*
$$\Phi^{-1}\left(\gamma \right)=\left\{\begin{array}{cc} a-b\sqrt{1-2\gamma }, & if\;\gamma \leq 0.5\\ a+c-c\sqrt{2-2\gamma }, & if\;\gamma >0.5. \end{array}\right.$$
*and then, it follows from Theorem 5 that the entropy of δ is:*
(20)$$\begin{array}{cc} \;H[\delta ]\;\;\;\; & =\int_{0}^{0.5}\left(a-b\sqrt{1-2x}\right)\ln\frac{x}{1-x}dx+\int_{0.5}^{1}\left(a+c-c\sqrt{2-2x}\right)\ln\frac{x}{1-x}dx\\ & =c\ln2-b\int_{0}^{1}u^{2}\ln\left(\frac{1-u^{2}}{1+u^{2}}\right)du+c\int_{0}^{1}u^{2}\ln\left(\frac{u^{2}}{2-u^{2}}\right)du\\ & =c\left(\ln2+\frac{4}{3}-\frac{4\sqrt{2}}{3}\ln\left(1+\sqrt{2}\right)\right)-b\left(\frac{\ln2}{3}-\frac{4}{3}+\frac{\pi }{6}\right). \end{array}$$

### 3.3. The Entropy of a Linear Function

Seeing that Liu [34] and Yao and Ke [35] have uncovered the evidence to support the linearity property of the entropy in uncertain sets, to verify the existence of this desirable property in the credibility-based entropy, the formula of calculating the entropy of a linear function constructed by regular LR fuzzy numbers is considered in this section and presented in the following form.

**Theorem** **6.***Suppose that $\delta_{1},\delta_{2},\cdots ,\delta_{n}$ are independent regular LR fuzzy numbers, and $f\left(x_{1},x_{2},\cdots ,x_{n}\right)=\lambda_{1}x_{1}+\lambda_{2}x_{2}+\cdots +\lambda_{n}x_{n}$, where $\lambda_{1},\lambda_{2},\cdots ,\lambda_{n}$ are real numbers. Then, the entropy of $\delta =f(\delta_{1},\delta_{2},\cdots ,\delta_{n})$ is:*
(21)$$H\left[\delta \right]=|\lambda_{1}|H\left[\delta_{1}\right]+|\lambda_{2}|H\left[\delta_{2}\right]+\cdots +\left|\lambda_{n}\right|H\left[\delta_{n}\right].$$

**Proof** **of Theorem 6.**For simplicity, we only prove the case of n=2 and $\delta =f(\delta_{1},\delta_{2})$. Denote the ICDs of δ, $\delta_{1}$, and $\delta_{2}$ by $\Phi^{-1}$, $\Phi_{1}^{-1}$, and $\Phi_{2}^{-1}$, respectively.Case (a): If $\lambda_{1}>0,\lambda_{2}>0$, then *f* is a strictly increasing function of $x_{1}$ and $x_{2}$. By Theorem 1, we have:
$$\Phi^{-1}\left(\gamma \right)=f\left(\Phi_{1}^{-1}\left(\gamma \right),\Phi_{2}^{-1}\left(\gamma \right)\right).$$
In view of Theorem 5, we can deduce that:
$$\begin{array}{cc} \;H[f\left(\delta_{1},\delta_{2}\right)]\;\;\;\; & =\int_{0}^{1}\left(\lambda_{1}\Phi_{1}^{-1}\left(\gamma \right)+\lambda_{2}\Phi_{2}^{-1}\left(\gamma \right)\right)\ln\frac{\gamma }{1-\gamma }d\gamma \\ & =\lambda_{1}\int_{0}^{1}\Phi_{1}^{-1}\left(\gamma \right)\ln\frac{\gamma }{1-\gamma }d\gamma +\lambda_{2}\int_{0}^{1}\Phi_{2}^{-1}\left(\gamma \right)\ln\frac{\gamma }{1-\gamma }d\gamma \\ & =\lambda_{1}H\left[\delta_{1}\right]+\lambda_{2}H\left[\delta_{2}\right]\\ & =|\lambda_{1}|H\left[\delta_{1}\right]+\left|\lambda_{2}\right|H\left[\delta_{2}\right]. \end{array}$$Case (b): If $\lambda_{1}>0,\lambda_{2}<0$, then *f* is strictly increasing with regard to $x_{1}$ and strictly decreasing with regard to $x_{2}$. On the basis of Theorem 1, we have:
$$\Phi^{-1}\left(\gamma \right)=f\left(\Phi_{1}^{-1}\left(\gamma \right),\Phi_{2}^{-1}\left(1-\gamma \right)\right).$$
According to Theorem 5, it can be derived that:
$$\begin{array}{cc} \;H[f\left(\delta_{1},\delta_{2}\right)]\;\;\;\; & =\int_{0}^{1}\left(\lambda_{1}\Phi_{1}^{-1}\left(\gamma \right)+\lambda_{2}\Phi_{2}^{-1}\left(1-\gamma \right)\right)\ln\frac{\gamma }{1-\gamma }d\gamma \\ & =\lambda_{1}\int_{0}^{1}\Phi_{1}^{-1}\left(\gamma \right)\ln\frac{\gamma }{1-\gamma }d\gamma +\lambda_{2}\int_{0}^{1}\Phi_{2}^{-1}\left(1-\gamma \right)\ln\frac{\gamma }{1-\gamma }d\gamma \\ & =\lambda_{1}\int_{0}^{1}\Phi_{1}^{-1}\left(\gamma \right)\ln\frac{\gamma }{1-\gamma }d\gamma -\lambda_{2}\int_{0}^{1}\Phi_{2}^{-1}\left(\gamma \right)\ln\frac{\gamma }{1-\gamma }d\gamma \\ & =\lambda_{1}H\left[\delta_{1}\right]-\lambda_{2}H\left[\delta_{2}\right]\\ & =|\lambda_{1}|H\left[\delta_{1}\right]+\left|\lambda_{2}\right|H\left[\delta_{2}\right]. \end{array}$$
Analogously, the same results can be reached for the other two cases (i.e., $\lambda_{1}<0$, $\lambda_{2}>0$ and $\lambda_{1}<0$, $\lambda_{2}<0$). In addition, if $\lambda_{1}=0$ or $\lambda_{2}=0$, the same results can be obtained easily. That is, for any real numbers $\lambda_{1}$ and $\lambda_{2}$, the equation $H[f\left(\delta_{1},\delta_{2}\right)]=|\lambda_{1}|H\left[\delta_{1}\right]+\left|\lambda_{2}\right|H\left[\delta_{2}\right]$ holds. □

By using Equation (Equation 21), the only thing that needs to be done for calculating the entropy of a linear function is calculating the entropy of each independent fuzzy number, whether the fuzzy numbers are of the same type or not.

**Example** **9.***Let $\delta_{1}∼T{\left(a_{1},b_{1},c_{1}\right)}_{LR}$ and $\delta_{2}∼T{\left(a_{2},b_{2},c_{2}\right)}_{LR}$ be two independent triangular fuzzy numbers, and $f\left(\delta_{1},\delta_{2}\right)=3\delta_{1}+5\delta_{2}$. In light of Equations (Equation 16) and (Equation 21), the entropy of $f(\delta_{1},\delta_{2})$ is:*
$$\begin{array}{cc} H[f\left(\delta_{1},\delta_{2}\right)]\;\;\;\; & =3H\left[\delta_{1}\right]+5H\left[\delta_{2}\right]=3\times \frac{b_{1}+c_{1}}{2}+5\times \frac{b_{2}+c_{2}}{2}=\frac{3b_{1}+3c_{1}+5b_{2}+5c_{2}}{2}. \end{array}$$

**Example** **10.***Let $\delta_{1}∼T{\left(a_{1},b_{1},c_{1}\right)}_{LR}$ be a triangular fuzzy number and $\delta_{2}∼M{\left(a_{2},b_{2},c_{2}\right)}_{LR}$ be a mixture fuzzy number, and $f\left(\delta_{1},\delta_{2}\right)=3\delta_{1}-5\delta_{2}$. In accordance with Theorem 6 and the results of Equations (Equation 16) and (Equation 20), the entropy of $f(\delta_{1},\delta_{2})$ is:*
$$\begin{array}{cc} H[f\left(\delta_{1},\delta_{2}\right)]\;\;\;\; & =3H\left[\delta_{1}\right]-5H\left[\delta_{2}\right]\\ & =3\times \frac{b_{1}+c_{1}}{2}+5\times \left(c_{2}\left(\ln2-\frac{4}{3}+\frac{4\sqrt{2}}{3}\ln\left(1+\sqrt{2}\right)\right)-b_{2}\left(\frac{ln2}{3}-\frac{4}{3}+\frac{\pi }{6}\right)\right)\\ & =\frac{1}{6}\left(9b_{1}+9c_{1}+10c_{2}\left(3\ln2-4+4\sqrt{2}\ln\left(1+\sqrt{2}\right)\right)-5b_{2}\left(2\ln2-8+\pi \right)\right). \end{array}$$

## 4. The Semi-Entropies of a Linear Function with LR Fuzzy Numbers

This section firstly gives the definitions of the lower semi-entropy proposed by Zhou et al. [39] and the upper semi-entropy suggested in the present work. After that, some formulas via the ICDs are proposed for calculating the lower and upper semi-entropies of a regular LR fuzzy number and a linear function with regular LR fuzzy numbers, and some examples are given.

### 4.1. The Definitions of Semi-Entropies

To quantify the downside uncertainty, Zhou et al. [39] put forward the concept of semi-entropy (hereinafter referred to as lower semi-entropy) in portfolio selection problems as follows.

**Definition** **7**(Zhou et al. [39])**.**
*Let δ be a continuous fuzzy number with expected value e and CF v. Then, its lower semi-entropy can be expressed as:*
(22)$$H_{S}{\left[\delta \right]}^{-}=\int_{-\infty }^{+\infty }S\left(v{\left(x\right)}^{-}\right)dx,$$
*where S(t)=−tlnt−(1−t)ln(1−t), and:*
(23)$$v{\left(x\right)}^{-}=\left\{\begin{array}{cc} v(x), & if\;x\leq e\\ 0, & if\;x>e. \end{array}\right.$$
*Since S(0)=0, the lower semi-entropy of the fuzzy number δ can be simplified as the following specification:*
(24)$$H_{S}{\left[\delta \right]}^{-}=\int_{-\infty }^{e}S\left(v\left(x\right)\right)dx.$$

While investors in finance focus their attention on the downside uncertainty, decision makers in engineering care more about the upside uncertainty that may increase the cost. On this account, a concept of upper semi-entropy is proposed in this paper.

**Definition** **8.***Suppose that δ is a continuous fuzzy number with expected value e and CF v. Then, its upper semi-entropy can be expressed as:*
(25)$$H_{S}{\left[\delta \right]}^{+}=\int_{-\infty }^{+\infty }S\left(v{\left(x\right)}^{+}\right)dx,$$
*where S(t)=−tlnt−(1−t)ln(1−t) and:*
(26)$$v{\left(x\right)}^{+}=\left\{\begin{array}{cc} 0, & if\;x\leq e\\ v(x), & if\;x>e. \end{array}\right.$$
*Since S(0)=0, the upper semi-entropy of the fuzzy number δ can be simplified as the following specification:*
(27)$$H_{S}{\left[\delta \right]}^{+}=\int_{e}^{+\infty }S\left(v\left(x\right)\right)dx.$$

**Corollary** **1.***Let δ be a continuous fuzzy number. We have:*
(28)$$H\left[\delta \right]=H_{S}{\left[\delta \right]}^{-}+H_{S}{\left[\delta \right]}^{+}.$$

**Proof** **of Corollary 1.**It follows from Definitions 6–8 and Equations (Equation 15), (Equation 24), and (Equation 27) immediately. □

**Example** **11.**Given that $\delta ∼T{\left(a,b,c\right)}_{LR}$ is a triangular fuzzy number in Example 1 and Figure 1a, then the expected value of δ can be deduced easily as $e=\frac{1}{4}\left(4a-b+c\right)$ by Theorem 3.*If e≤a, in view of Equations (Equation 9), (Equation 16), (Equation 24), and (Equation 28), we have:*
$$\begin{array}{c} \;H_{S}{\left[\delta \right]}^{-}=\int_{a-b}^{e}S\left(\frac{1}{2}\left(1-\frac{a-x}{b}\right)\right)dx=\int_{0}^{m}2bS\left(t\right)dt=bT\left(m\right),\\ \;H_{S}{\left[\delta \right]}^{+}=H\left[\delta \right]-H_{S}{\left[\delta \right]}^{-}=\frac{b+c-2bT(m)}{2}, \end{array}$$
*where $T\left(x\right)={\left(1-x\right)}^{2}\ln\left(1-x\right)-x^{2}\ln x+x$ and $m=\frac{3b\;+\;c}{8b}$.**If e>a, it can be figured out that:*
$$\begin{array}{c} \;H_{S}{\left[\delta \right]}^{+}=\int_{e}^{a+c}S\left(\frac{1}{2}\left(1-\frac{x-a}{c}\right)\right)dx=\int_{0}^{n}2cS\left(t\right)dt=cT\left(n\right),\\ \;H_{S}{\left[\delta \right]}^{-}=H\left[\delta \right]-H_{S}{\left[\delta \right]}^{+}=\frac{b+c-2cT(n)}{2}, \end{array}$$
*where $T\left(x\right)={\left(1-x\right)}^{2}\ln\left(1-x\right)-x^{2}\ln x+x$ and $n=\frac{b+3c}{8c}$.*

### 4.2. The Semi-Entropy of a Regular LR Fuzzy Number

Similar to the entropy of a regular LR fuzzy number, the lower and upper semi-entropies can be also derived via the CD and ICD.

**Theorem** **7.***Let δ be a continuous fuzzy number with CD *Φ*. Then, its lower and upper semi-entropies can be calculated, respectively, as:*
(29)$$\;H_{S}{\left[\delta \right]}^{-}=\int_{-\infty }^{e}S\left(\Phi \left(x\right)\right)dx,$$
(30)$$H_{S}{\left[\delta \right]}^{+}=\int_{e}^{+\infty }S\left(\Phi \left(x\right)\right)dx.$$

**Proof** **of Theorem 7.**Assume that e,σ,v are the expected value, mean value, and CF of a continuous fuzzy number δ, respectively.If e≤σ, on the basis of Equations (Equation 18), (Equation 22), and (Equation 23), we can deduce that:
$$H_{S}{\left[\delta \right]}^{-}\;=\int_{-\infty }^{e}S\left(v(x)\right)dx=\int_{-\infty }^{e}S\left(\Phi (x)\right)dx.$$Similarly, if e>σ, we have:
$$\begin{array}{cc} \;H_{S}{\left[\delta \right]}^{-}\;\;\; & \;\;=\int_{-\infty }^{\sigma }S\left(v(x)\right)dx+\int_{\sigma }^{e}S\left(1-v(x)\right)dx=\int_{-\infty }^{e}S\left(\Phi \left(x\right)\right)dx. \end{array}$$The formula for the upper semi-entropy in Equation (Equation 30) can be proven in a similar way. □

**Theorem** **8.***Let δ be a regular LR fuzzy number with ICD $\Phi^{-1}$ and expected value e. Then, the lower and upper semi-entropies of δ can be calculated as:*
(31)$$H_{S}{\left[\delta \right]}^{-}=\left\{\begin{array}{cc} \int_{0}^{\Phi (e)}(\Phi^{-1}\left(\gamma )-e\right)\ln\frac{\gamma }{1-\gamma }d\gamma , & if\;e\leq 0\\ \int_{0}^{\Phi (e)}\Phi^{-1}\left(\gamma \right)\ln\frac{\gamma }{1-\gamma }d\gamma +e\int_{\Phi (e)}^{1}\ln\frac{\gamma }{1-\gamma }d\gamma , & if\;e>0, \end{array}\right.$$
(32)$$H_{S}{\left[\delta \right]}^{+}=\left\{\begin{array}{cc} e\int_{0}^{\Phi (e)}\ln\frac{\gamma }{1-\gamma }d\gamma +\int_{\Phi (e)}^{1}\Phi^{-1}\left(\gamma \right)\ln\frac{\gamma }{1-\gamma }d\gamma , & if\;e\leq 0\\ \int_{\Phi (e)}^{1}(\Phi^{-1}\left(\gamma )-e\right)\ln\frac{\gamma }{1-\gamma }d\gamma , & if\;e>0. \end{array}\right.$$

**Proof** **of Theorem 8.**Following Theorem 7, the lower semi-entropy of δ is:
$$H_{S}{\left[\delta \right]}^{-}=\int_{-\infty }^{e}S\left(v\left(x\right)\right)dx=\int_{-\infty }^{e}S\left(\Phi \left(x\right)\right)dx.$$If e≤0, by the Fubini theorem, we can further obtain that:
$$\begin{array}{cc} \;H_{S}{\left[\delta \right]}^{-}\;\; & \;\;\;=\int_{-\infty }^{e}\int_{0}^{\Phi (x)}S^{\prime }\left(\gamma \right)d\gamma dx=\int_{0}^{\Phi (e)}\int_{\Phi^{-1}\left(\gamma \right)}^{e}S^{\prime }\left(\gamma \right)dxd\gamma \\ & =\int_{0}^{\Phi (e)}\left(\Phi^{-1}\left(\gamma \right)-e\right)\ln\frac{\gamma }{1-\gamma }d\gamma . \end{array}$$If e>0, we have:
$$\begin{array}{cc} H_{S}{\left[\delta \right]}^{-}\;\; & \;\;\;=\int_{-\infty }^{0}\int_{0}^{\Phi (x)}S^{\prime }\left(\gamma \right)d\gamma dx+\int_{0}^{e}\int_{\Phi (x)}^{1}-S^{\prime }\left(\gamma \right)d\gamma dx\\ & =\int_{0}^{\Phi (0)}\int_{\Phi^{-1}\left(\gamma \right)}^{0}S^{\prime }\left(\gamma \right)dxd\gamma +\int_{\Phi (0)}^{\Phi (e)}\int_{0}^{\Phi^{-1}\left(\gamma \right)}-S^{\prime }\left(\gamma \right)dxd\gamma +\int_{\Phi (e)}^{1}\int_{0}^{e}-S^{\prime }\left(\gamma \right)dxd\gamma \\ & =\int_{0}^{\Phi (e)}\Phi^{-1}\left(\gamma \right)\ln\frac{\gamma }{1-\gamma }d\gamma +e\int_{\Phi (e)}^{1}\ln\frac{\gamma }{1-\gamma }d\gamma . \end{array}$$A similar procedure could be adapted easily to deduce Equation (Equation 32). □

Similar to Theorem 5, Theorem 8 gives a formula for calculating the semi-entropies with relatively less complexity compared with Definitions 7 and 8 and Theorem 7.

**Example** **12.**Assume that $\delta ∼N{\left(1,2,3\right)}_{LR}$ is a normal fuzzy number in Example 3 and Figure 1c. Then, the expected value of δ is $e=\frac{1}{6}\left(6a-b+c\right)=\frac{7}{6}$ by Theorem 3.*According to Equation (Equation 10) and Theorem 8, e>0 and $\Phi \left(e\right)=\frac{19}{36}>0.5$, we have:*
$$\begin{array}{cc} \;H_{S}{\left[\delta \right]}^{-} & \;=\int_{0}^{0.5}\left(a-b+b\sqrt{2x}\right)\ln\frac{x}{1-x}dx+\int_{0.5}^{\Phi (e)}\left(a+c-c\sqrt{2-2x}\right)\ln\frac{x}{1-x}dx+e\int_{\Phi (e)}^{1}\ln\frac{x}{1-x}dx\\ & \;=\int_{0}^{0.5}\left(2\sqrt{2x}-1\right)\ln\frac{x}{1-x}dx+\int_{0.5}^{\frac{19}{36}}\left(4-3\sqrt{2-2x}\right)\ln\frac{x}{1-x}dx+\frac{7}{6}\int_{\frac{19}{36}}^{1}\ln\frac{x}{1-x}dx\\ & \;=0.844,\\ \;H_{S}{\left[\delta \right]}^{+} & \;=\int_{\Phi (e)}^{1}\left(a+c-c\sqrt{2-2x}-e\right)\ln\frac{x}{1-x}dx=\int_{\frac{19}{36}}^{1}\left(\frac{17}{6}-3\sqrt{2-2x}\right)\ln\frac{x}{1-x}dx=0.978. \end{array}$$

### 4.3. The Semi-Entropies of a Linear Function

**Theorem** **9.***Let $\delta_{1}$, $\delta_{2}$, ⋯, $\delta_{n}$ be independent regular LR fuzzy numbers with ICDs $\Phi_{1}^{-1},\Phi_{2}^{-1},\cdots ,\Phi_{n}^{-1}$, respectively, and $\lambda_{1},\lambda_{2},\cdots ,\lambda_{m}\geq 0$, $\lambda_{m+1},\lambda_{m+2},\cdots ,\lambda_{n}\leq 0$ be real numbers. Then, the lower and upper semi-entropies of $\delta =\lambda_{1}\delta_{1}+\lambda_{2}\delta_{2}+\cdots +\lambda_{n}\delta_{n}$ can be calculated as Equations (Equation 31) and (Equation 32), respectively, where the ICD of δ is:*
(33)$$\Phi^{-1}\left(\gamma \right)=\lambda_{1}\Phi_{1}^{-1}\left(\gamma \right)+\cdots +\lambda_{m}\Phi_{m}^{-1}\left(\gamma \right)+\lambda_{m+1}\Phi_{m+1}^{-1}\left(1-\gamma \right)+\cdots +\lambda_{n}\Phi_{n}^{-1}\left(1-\gamma \right),$$
*and the expected value of δ is $e=\int_{0}^{1}\Phi^{-1}\left(\gamma \right)d\gamma$.*

**Proof** **of Theorem 9.**It follows from Theorems 1 and 8 immediately. □

**Example** **13.***Let $\delta_{1}∼T{\left(1,2,3\right)}_{LR}$ be a triangular fuzzy number and $\delta_{2}∼N{\left(1,2,3\right)}_{LR}$ be a normal fuzzy number. In view of Equation (Equation 10), the ICDs of $\delta_{1}$ and $\delta_{2}$ can be derived, respectively, as:*
$$\Phi_{1}^{-1}\left(\gamma \right)=\left\{\begin{array}{cc} 4\gamma -1, & if\;\gamma \leq 0.5\\ 6\gamma -2, & if\;\gamma >0.5, \end{array}\right.\Phi_{2}^{-1}\left(\gamma \right)=\left\{\begin{array}{cc} 2\sqrt{2\gamma }-1, & if\;\gamma \leq 0.5\\ 4-3\sqrt{2-2\gamma }, & if\;\gamma >0.5. \end{array}\right.$$
*The ICD of $\delta =\delta_{1}-\delta_{2}$ is:*
(34)$$\Phi^{-1}\left(\gamma \right)=\Phi_{1}^{-1}\left(\gamma \right)-\Phi_{2}^{-1}\left(1-\gamma \right)=\left\{\begin{array}{cc} 4\gamma +3\sqrt{2\gamma }-5, & if\;\gamma \leq 0.5\\ 6\gamma -2\sqrt{2-2\gamma }-1, & if\;\gamma >0.5. \end{array}\right.$$
*According to Theorem 3, the expected value of δ is:*
(35)$$e=\int_{0}^{1}\Phi^{-1}\left(\gamma \right)d\gamma =\frac{1}{12}.$$
*By Equations (Equation 34) and (Equation 35), Φ(e)=0.512. On the basis of Equations (Equation 31) and (Equation 32), the lower semi-entropy of δ is:*
$$\begin{array}{cc} \;H_{S}{\left[\delta \right]}^{-}\;\;\;\;\; & =\int_{0}^{\Phi (e)}\Phi^{-1}\left(\gamma \right)\ln\frac{\gamma }{1-\gamma }d\gamma +e\int_{\Phi (e)}^{1}\ln\frac{\gamma }{1-\gamma }d\gamma \\ & =\int_{0}^{0.5}\left(4\gamma +3\sqrt{2\gamma }-5\right)\ln\frac{\gamma }{1-\gamma }d\gamma +\int_{0.5}^{0.512}\left(6\gamma -2\sqrt{2-2\gamma }-1\right)\ln\frac{\gamma }{1-\gamma }d\gamma \\ & \;\;\;+\frac{1}{12}\int_{0.512}^{1}\ln\frac{\gamma }{1-\gamma }d\gamma \\ & =2.1513, \end{array}$$
*and the upper semi-entropy of fδ is:*
$$\begin{array}{cc} \;H_{S}{\left[\delta \right]}^{+}\;\;\;\;\; & =\int_{\Phi (e)}^{1}\left(\Phi^{-1}\left(\gamma \right)-e\right)\ln\frac{\gamma }{1-\gamma }d\gamma =\int_{0.512}^{1}\left(6\gamma -2\sqrt{2-2\gamma }-\frac{13}{12}\right)\ln\frac{\gamma }{1-\gamma }d\gamma =2.1713. \end{array}$$

In the cases where $\delta_{1}$, $\delta_{2}$, ⋯, $\delta_{n}$ are of the same type, that is they have the same shape functions *L* and *R*, but with different parameter settings $(a_{i},b_{i},c_{i}),i=1,2,\cdots ,n$, it is well-known that $\delta =\lambda_{1}\delta_{1}+\lambda_{2}\delta_{2}+\cdots +\lambda_{n}\delta_{n}$ is also a fuzzy number with the same shape functions, and the parameters *a*, *b*, and *c* can be deduced by $a_{i}$, $b_{i}$, $c_{i}$, and $\lambda_{i}$. For example, when $\lambda_{i}\geq 0$, we have:$$a=\sum_{i=1}^{n}\lambda_{i}a_{i},\;b=\sum_{i=1}^{n}\lambda_{i}b_{i},\;c=\sum_{i=1}^{n}\lambda_{i}c_{i}.$$
Thereafter, the semi-entropies of δ can be calculated directly by Equations (Equation 31) and (Equation 32). This method can reduce calculation load greatly especially when the size *n* is large.

**Example** **14.***Let $\delta_{1}∼T{\left(1,2,1\right)}_{LR}$, $\delta_{2}∼T{\left(3,1,2\right)}_{LR}$, $\delta_{3}∼T{\left(-2,2,1\right)}_{LR}$, and $\delta_{4}∼T{\left(1,2,2\right)}_{LR}$ be four independent triangular fuzzy numbers. Then, $\delta =\delta_{1}+\delta_{2}+\delta_{3}+\delta_{4}$ is still a triangular fuzzy number with $\delta ∼T{\left(3,7,6\right)}_{LR}$. Consequently, in light of Theorem 3 and Equation (Equation 10), the expected value of δ is $e=\frac{11}{4}$, and its ICD is:*
$$\Phi^{-1}\left(\gamma \right)=\left\{\begin{array}{cc} 14\gamma -4, & if\;\gamma \leq 0.5\\ 12\gamma -3, & if\;\gamma >0.5. \end{array}\right.$$
*Accordingly, we have $\Phi \left(e\right)=\frac{23}{48}$. By Equations (Equation 31) and (Equation 32), the lower semi-entropy of δ is:*
$$\begin{array}{cc} \;H_{S}{\left[\delta \right]}^{-}\;\;\;\;\; & =\int_{0}^{\Phi (e)}\Phi^{-1}\left(\gamma \right)\ln\frac{\gamma }{1-\gamma }d\gamma +e\int_{\Phi (e)}^{1}\ln\frac{\gamma }{1-\gamma }d\gamma \\ & =\int_{0}^{\frac{23}{48}}\left(14\gamma -4\right)\ln\frac{\gamma }{1-\gamma }d\gamma +\frac{11}{4}\int_{\frac{23}{48}}^{1}\ln\frac{\gamma }{1-\gamma }d\gamma \\ & =3.3268, \end{array}$$
*and the upper semi-entropy of δ is:*
$$\begin{array}{cc} \;H_{S}{\left[\delta \right]}^{+}\;\;\;\;\; & =\int_{\Phi (e)}^{1}\left(\Phi^{-1}\left(\gamma \right)-e\right)\ln\frac{\gamma }{1-\gamma }d\gamma \\ & =\int_{\frac{23}{48}}^{0.5}\left(14\gamma -\frac{27}{4}\right)\ln\frac{\gamma }{1-\gamma }d\gamma +\int_{0.5}^{1}\left(12\gamma -\frac{23}{4}\right)\ln\frac{\gamma }{1-\gamma }d\gamma \\ & =3.1733. \end{array}$$

## 5. Entropy Optimization Models

Fuzzy programming (FP), which aims to find optimal solutions under a set of linear or nonlinear objectives and constraints involving fuzzy parameters or variables, is an important optimization technique in both theory and application. Some indicators such as mean and variance are employed to characterize and defuzzify the models. In this section, the entropy and semi-entropies are used in FP models as alternative measures of characterizing the uncertainty with regard to return and cost-oriented problems.

### 5.1. Mean-Entropy and Mean-Semi-Entropy Optimization Models in Return-Oriented Problems

For a decision maker with a given input, achieving more returns at lower risks is the main purpose while making decisions among multiple alternatives. Generally, the decision maker has an expectation of the return, but little knowledge of predicting the risk due to information vagueness and imprecision. Using fuzzy parameters to express the returns of the alternatives and entropy to denote the risks, the general paradigm of a mean-entropy optimization model aiming to minimize the risk under certain expected constraints can be described as follows,
(36)$$\left\{\begin{array}{c} \min H[f(x,\delta )]\\ \mathrm{subject}\;\mathrm{to}:\\ \;\;\;\;\;\;\;\;\;\;\;E[f(x,\delta )]\geq \alpha \\ \;\;\;\;\;\;\;\;\;\;\;E[g_{i}\left(x,\delta \right)]\leq 0,\;i=1,2,\cdots ,n, \end{array}\right.$$
where f(x,δ) is the overall return with $x=(x_{1},x_{2},\cdots ,x_{p})$ as decision variables and $\delta =(\delta_{1},\delta_{2},\cdots ,$$\delta_{m})$ as fuzzy parameters, $g_{i}\left(x,\delta \right),i=1,2,\cdots ,n,$ are the constraint functions, and α is a predetermined constant denoting the lower limit of the expected overall return.

Considering that entropy measures bilateral risks, minimizing the entropy will certainly sacrifice high returns. For a decision maker who focuses on the downside risk only and is willing to get high returns, a mean-semi-entropy model with the objective to minimize the lower semi-entropy is more appropriate. The model satisfying certain expected return constraints is shown below,
(37)$$\left\{\begin{array}{c} \min H_{S}{\left[f\left(x,\delta \right)\right]}^{-}\\ \mathrm{subject}\;\mathrm{to}:\\ \;\;\;\;\;\;\;\;\;\;\;E[f(x,\delta )]\geq \alpha \\ \;\;\;\;\;\;\;\;\;\;\;E[g_{i}\left(x,\delta \right)]\leq 0,\;i=1,2,\cdots ,n, \end{array}\right.$$
where $H_{S}{\left[f\left(x,\delta \right)\right]}^{-}$ is the lower semi-entropy of the overall return.

In particular, if *f* and $g_{i},i=1,2,\cdots ,n,$ are linear functions of independent and regular fuzzy parameters $\delta_{k}$, i.e.,
$$f\left(x,\delta \right)=f_{0}\left(x\right)+\sum_{k=1}^{m}f_{k}\left(x\right)\delta_{k},\;\;\;g_{i}\left(x,\delta \right)=g_{i0}\left(x\right)+\sum_{k=1}^{m}g_{ik}\left(x\right)\delta_{k},$$
then Model (Equation 36) can be transformed into the following form based on Theorems 2 and 6,
(38)$$\left\{\begin{array}{c} \min H\left[\delta_{1}\right]|f_{1}\left(x\right)|+H\left[\delta_{2}\right]|f_{2}\left(x\right)|+\cdots +H\left[\delta_{m}\right]\left|f_{m}\left(x\right)\right|\\ \mathrm{subject}\;\mathrm{to}:\\ \;\;\;\;\;\;\;\;\;\;\;f_{0}\left(x\right)+E\left[\delta_{1}\right]f_{1}\left(x\right)+E\left[\delta_{2}\right]f_{2}\left(x\right)+\cdots +E\left[\delta_{m}\right]f_{m}\left(x\right)\geq \alpha \\ \;\;\;\;\;\;\;\;\;\;\;g_{i0}\left(x\right)+E\left[\delta_{1}\right]g_{i1}\left(x\right)+E\left[\delta_{2}\right]g_{i2}\left(x\right)+\cdots +E\left[\delta_{m}\right]g_{im}\left(x\right)\leq 0,\;i=1,2,\cdots ,n, \end{array}\right.$$
where $H[\delta_{k}]$ and $E[\delta_{k}]$ are the entropies and expectations of $\delta_{k},k=1,2,\cdots ,m,$ respectively. Since the entropy of a constant is 0, $f_{0}\left(x\right)$ is eliminated in the objective function.

Likewise, the equivalent model of Model (Equation 37) by Theorem 9 can be depicted as:(39)$$\left\{\begin{array}{c} \min H_{S}{\left[f\left(x,\delta \right)\right]}^{-}\\ \mathrm{subject}\;\mathrm{to}:\\ \;\;\;\;\;\;\;\;\;\;\;f_{0}\left(x\right)+E\left[\delta_{1}\right]f_{1}\left(x\right)+E\left[\delta_{2}\right]f_{2}\left(x\right)+\cdots +E\left[\delta_{m}\right]f_{m}\left(x\right)\geq \alpha \\ \;\;\;\;\;\;\;\;\;\;\;g_{i0}\left(x\right)+E\left[\delta_{1}\right]g_{i1}\left(x\right)+E\left[\delta_{2}\right]g_{i2}\left(x\right)+\cdots +E\left[\delta_{m}\right]g_{im}\left(x\right)\leq 0,\;i=1,2,\cdots ,n, \end{array}\right.$$
where $H_{S}{\left[f\left(x,\delta \right)\right]}^{-}$ herein is an integral expression of $f_{k}\left(x\right)$. When the CDs $\Phi_{k}$ and the expected values $e_{k}$ of $\delta_{k}$ are known, it can be expressed as Equation (Equation 31).

### 5.2. Mean-Entropy and Mean-Semi-Entropy Optimization Models in Cost-Oriented Problems

For cost-oriented problems under uncertain environments, a decision maker usually wants to keep the costs within reasonable boundaries with the risks controllable rather than minimize the costs for unpredictable risks. In such a situation, using fuzzy parameters to express the costs of the choices and entropy to measure the risks, then the general paradigm of a mean-entropy optimization model with the objective of entropy minimization under certain expected constraints is:(40)$$\left\{\begin{array}{c} \min H[f(x,\delta )]\\ \mathrm{subject}\;\mathrm{to}:\\ \;\;\;\;\;\;\;\;\;\;\;E[f(x,\delta )]\leq \beta \\ \;\;\;\;\;\;\;\;\;\;\;E[g_{i}\left(x,\delta \right)]\geq 0,\;i=1,2,\cdots ,n, \end{array}\right.$$
where f(x,δ) is the overall cost, $g_{i}\left(x,\delta \right)$ are constraint functions, and β is a predetermined constant denoting the upper limits of the expected cost.

Compared with the lower semi-entropy measuring the downside risk that the return is less than the expected value, the upper semi-entropy weighing the upside risk that the cost exceeds the expected value is more applicable in this case. Thus, the mean-semi-entropy optimization model of minimizing the upper semi-entropy can be established to handle cost-oriented problems as follows,
(41)$$\left\{\begin{array}{c} \min H_{S}{\left[f\left(x,\delta \right)\right]}^{+}\\ \mathrm{subject}\;\mathrm{to}:\\ \;\;\;\;\;\;\;\;\;\;\;E[f(x,\delta )]\leq \beta \\ \;\;\;\;\;\;\;\;\;\;\;E[g_{i}\left(x,\delta \right)]\geq 0,\;i=1,2,\cdots ,n, \end{array}\right.$$
where $H_{S}{\left[f\left(x,\delta \right)\right]}^{+}$ is the upper semi-entropy of the overall cost.

Similarly, when f(x,δ) and $g_{i}\left(x,\delta \right)$ are linear with respect to fuzzy parameters $\delta_{k}$, Models (Equation 40) and (Equation 41) can be transformed into:(42)$$\left\{\begin{array}{c} \min H\left[\delta_{1}\right]|f_{1}\left(x\right)|+H\left[\delta_{2}\right]|f_{2}\left(x\right)|+\cdots +H\left[\delta_{m}\right]\left|f_{m}\left(x\right)\right|\\ \mathrm{subject}\;\mathrm{to}:\\ \;\;\;\;\;\;\;\;\;\;\;f_{0}\left(x\right)+E\left[\delta_{1}\right]f_{1}\left(x\right)+E\left[\delta_{2}\right]f_{2}\left(x\right)+\cdots +E\left[\delta_{m}\right]f_{m}\left(x\right)\leq \beta \\ \;\;\;\;\;\;\;\;\;\;\;g_{i0}\left(x\right)+E\left[\delta_{1}\right]g_{i1}\left(x\right)+E\left[\delta_{2}\right]g_{i2}\left(x\right)+\cdots +E\left[\delta_{m}\right]g_{im}\left(x\right)\geq 0,\;i=1,2,\cdots ,n \end{array}\right.$$
and:(43)$$\left\{\begin{array}{c} \min H_{S}{\left[f\left(x,\delta \right)\right]}^{+}\\ \mathrm{subject}\;\mathrm{to}:\\ \;\;\;\;\;\;\;\;\;\;\;f_{0}\left(x\right)+E\left[\delta_{1}\right]f_{1}\left(x\right)+E\left[\delta_{2}\right]f_{2}\left(x\right)+\cdots +E\left[\delta_{m}\right]f_{m}\left(x\right)\leq \beta \\ \;\;\;\;\;\;\;\;\;\;\;g_{i0}\left(x\right)+E\left[\delta_{1}\right]g_{i1}\left(x\right)+E\left[\delta_{2}\right]g_{i2}\left(x\right)+\cdots +E\left[\delta_{m}\right]g_{im}\left(x\right)\geq 0,\;i=1,2,\cdots ,n, \end{array}\right.$$
where $H[\delta_{k}]$ and $E[\delta_{k}]$ are the entropies and expectations of $\delta_{k},k=1,2,\cdots ,m$, respectively, and $H_{S}{\left[f\left(x,\delta \right)\right]}^{+}$ herein is an integral expression of $f_{k}\left(x\right)$ like Equation (Equation 32) when $\Phi_{k}$ and $e_{k}$ are known.

In brief, when the objective and constraints are linear functions of the fuzzy parameters, the generalized FP models above can be transformed into their equivalent forms like Model (Equation 38), (Equation 39), (Equation 42), or (Equation 43). Since the CDs $\Phi_{i}$ of the fuzzy parameters tend to be known in practice, the expectations $E[\delta_{i}]$ and entropies $H[\delta_{i}]$ (semi-entropies $H_{S}{\left[f\right]}^{-}$, $H_{S}{\left[f\right]}^{+}$) in these models can be derived by the proposed calculation formulas in advance, and then, the models turn into crisp linear or nonlinear programming models not including uncertain operators and can be solved by classical mathematical optimization methods or algorithms readily with the help of software.

### 5.3. Numerical Examples

In this section, a portfolio selection problem and a project selection problem are illustrated to show the solution framework of the two types of entropy optimization models, respectively. The results are solved by MATLAB 2015a.

**Example** **15.**This example is illustrated to compare the performance and differences of the mean-entropy and mean-semi-entropy optimization models for return-oriented problems, where the fuzzy parameters $\delta_{i}$ are triangular fuzzy numbers.

Suppose that an investor would like to build a portfolio from four stocks in which Stock 1 is a junk equity with large downside risks, Stock 2 is a plain and stable equity, Stock 3 is a blue chip, and Stock 4 is a growth equity. Denote by $\delta_{i}$ the fuzzy return of stock *i*, and the parameters of the four stocks are presented in Table 1, in which the expected values and entropies are calculated by Equations (Equation 13) and (Equation 16).

Denote by *a* the lower limit of the expected overall return. In this example, the investor expects to minimize the uncertainty of obtaining a return greater than *a*. Thus, the entropy optimization models in light of Models (Equation 36) and (Equation 37) are:$$\left\{\begin{array}{c} \min H[\delta_{1}x_{1}+\delta_{2}x_{2}+\delta_{3}x_{3}+\delta_{4}x_{4}]\\ \mathrm{subject}\;\mathrm{to}:\\ \;\;\;\;\;\;\;\;\;\;\;E[\delta_{1}x_{1}+\delta_{2}x_{2}+\delta_{3}x_{3}+\delta_{4}x_{4}]\geq a\\ \;\;\;\;\;\;\;\;\;\;\;x_{1}+x_{2}+x_{3}+x_{4}=1\\ \;\;\;\;\;\;\;\;\;\;\;x_{i}\geq 0,\;i=1,2,3,4 \end{array}\right.\mathrm{and}:\;\;\;\;\;\;\;\;\;\;\left\{\begin{array}{c} \min H_{S}{\left[\delta_{1}x_{1}+\delta_{2}x_{2}+\delta_{3}x_{3}+\delta_{4}x_{4}\right]}^{-}\\ \mathrm{subject}\;\mathrm{to}:\\ \;\;\;\;\;\;\;\;\;\;\;E[\delta_{1}x_{1}+\delta_{2}x_{2}+\delta_{3}x_{3}+\delta_{4}x_{4}]\geq a\\ \;\;\;\;\;\;\;\;\;\;\;x_{1}+x_{2}+x_{3}+x_{4}=1\\ \;\;\;\;\;\;\;\;\;\;\;x_{i}\geq 0,\;i=1,2,3,4, \end{array}\right.$$
respectively, where $x_{i}$ is the proportion allocated to stock *i*.

Since the overall return $\delta_{1}x_{1}+\delta_{2}x_{2}+\delta_{3}x_{3}+\delta_{4}x_{4}$ is linear with respect to $\delta_{i},i=1,2,3,4$, the equivalent formulations according to Models (Equation 38) and (Equation 39) and Table 1 are:$$\left\{\begin{array}{c} \min3.5x_{1}+2x_{2}+3.5x_{3}+6x_{4}\\ \mathrm{subject}\;\mathrm{to}:\\ \;\;\;\;\;\;\;\;\;\;\;-1.75x_{1}+2.5x_{2}+3.75x_{3}+3x_{4}\geq a\\ \;\;\;\;\;\;\;\;\;\;\;x_{1}+x_{2}+x_{3}+x_{4}=1\\ \;\;\;\;\;\;\;\;\;\;\;x_{i}\geq 0,\;i=1,2,3,4 \end{array}\right.\;\;\;\;\mathrm{and}:\;\;\;\;\;\;\;\left\{\begin{array}{c} \min H_{S}{\left[\delta_{1}x_{1}+\delta_{2}x_{2}+\delta_{3}x_{3}+\delta_{4}x_{4}\right]}^{-}\\ \mathrm{subject}\;\mathrm{to}:\\ \;\;\;\;\;\;\;\;\;\;\;-1.75x_{1}+2.5x_{2}+3.75x_{3}+3x_{4}\geq a\\ \;\;\;\;\;\;\;\;\;\;\;x_{1}+x_{2}+x_{3}+x_{4}=1\\ \;\;\;\;\;\;\;\;\;\;\;x_{i}\geq 0,\;i=1,2,3,4. \end{array}\right.$$

Obviously, the mean-entropy model on the left is a linear programming, which can be precisely solved by a software package. The mean-semi-entropy model on the right is a nonlinear programming in which the objective can be explicitly expressed by Equation (Equation 31) since $\delta_{1}x_{1}+\delta_{2}x_{2}+\delta_{3}x_{3}+\delta_{4}x_{4}$ is also a triangular fuzzy number whose parameters are linear functions of $x_{i},i=1,2,3,4$, and therefore, it can be easily solved by the MATLAB toolbox.

With the help of MATLAB 2015a, the optimal portfolios under the mean-entropy and mean-semi-entropy models with different lower limits of the expected overall returns *a* can be obtained as shown in Table 2 and Figure 3. It can be seen from the left side of Table 2 that the coefficients of $x_{1}$ and $x_{4}$ are all 0, i.e., only stocks 2 and 3 are included in the optimal portfolio, and the lower limit of the expected overall return merely influences the proportion allocated to them. The results can well explain that entropy represents the uncertainty of both high and low extreme returns, thereby the mean-entropy optimization model excludes Stock 1, which has the potential to lose 60%, and Stock 4, which can potentially reach a return of 90% simultaneously.

As shown in the right side of Table 2 and Figure 3, the optimal portfolios under the semi-entropy optimization model are more distributed, since Stock 4 is included in the portfolios and the proportions allocated to it are more than 20%. Besides, the higher return the investor expects, the higher the proportion allocated to Stock 4 is, indicating that investors who expect higher returns are willing to take higher risks and buy fluctuating stocks. Nevertheless, Stocks 2 and 3 still occupy the majority of the investments, and the coefficients of $x_{1}$ are still zero, which is in line with the actual decision-making mode that investors tend to invest large amounts of money in stable assets and reserve a small percentage for risky investments except for junk stocks, which may cause heavy losses, but gain little profits, then realize steady appreciation. The results imply that using semi-entropy as a risk measure can perfectly get rid of junk stocks and does not miss potentially high-return stocks. What is more, owing to the higher expected return of Stock 4, the expected return of the portfolio obtained by the mean-semi-entropy model is higher than that of the mean-entropy model at each lower limit. Note that since entropy measures bilateral uncertainty, yet lower semi-entropy weighs downside risks, the entropies are naturally larger than the semi-entropies, as shown in Table 2.

**Example** **16.**This example is used to illustrate the entropy and semi-entropy optimization models of cost-oriented problems, where the fuzzy parameters $\delta_{i}$ are triangular fuzzy numbers.

Consider a company that plans to develop some new products. There are four products whose R&D costs are supposed to be triangular fuzzy numbers. Let $x_{i}$ denote the proportion allocated to product *i* for i=1,2,3,4, respectively, $\delta_{i}$ the fuzzy cost of developing product *i*, and *b* the upper limit of the expected total cost. The parameters of the four products are represented in Table 3, in which the expected values and entropies are calculated by Equations (Equation 13) and (Equation 16).

In this example, if the company hopes to pay a total cost less than the upper limit *b* and minimize the uncertainty of the expenditure simultaneously, then the entropy optimization models are:$$\left\{\begin{array}{c} \min H[\delta_{1}x_{1}+\delta_{2}x_{2}+\delta_{3}x_{3}+\delta_{4}x_{4}]\\ \mathrm{subject}\;\mathrm{to}:\\ \;\;\;\;\;\;\;\;\;\;\;E[\delta_{1}x_{1}+\delta_{2}x_{2}+\delta_{3}x_{3}+\delta_{4}x_{4}]\leq b\\ \;\;\;\;\;\;\;\;\;\;\;x_{1}+x_{2}+x_{3}+x_{4}=1\\ \;\;\;\;\;\;\;\;\;\;\;x_{i}\geq 0,\;i=1,2,3,4 \end{array}\right.\;\;\;\mathrm{and}:\;\;\;\;\;\;\;\;\;\;\left\{\begin{array}{c} \min H_{S}{\left[\delta_{1}x_{1}+\delta_{2}x_{2}+\delta_{3}x_{3}+\delta_{4}x_{4}\right]}^{-}\\ \mathrm{subject}\;\mathrm{to}:\\ \;\;\;\;\;\;\;\;\;\;\;E[\delta_{1}x_{1}+\delta_{2}x_{2}+\delta_{3}x_{3}+\delta_{4}x_{4}]\leq b\\ \;\;\;\;\;\;\;\;\;\;\;x_{1}+x_{2}+x_{3}+x_{4}=1\\ \;\;\;\;\;\;\;\;\;\;\;x_{i}\geq 0,\;i=1,2,3,4. \end{array}\right.$$

Further, according to Models (Equation 42) and (Equation 43) and Table 3, the equivalent formulations of the above models are:$$\left\{\begin{array}{c} \min3.5x_{1}+2x_{2}+5x_{3}+5.5x_{4}\\ \mathrm{subject}\;\mathrm{to}:\\ \;\;\;\;\;\;\;\;\;\;\;7.25x_{1}+8.5x_{2}+11.5x_{3}+7.75x_{4}\leq b\\ \;\;\;\;\;\;\;\;\;\;\;x_{1}+x_{2}+x_{3}+x_{4}=1\\ \;\;\;\;\;\;\;\;\;\;\;x_{i}\geq 0,\;i=1,2,3,4 \end{array}\right.\;\;\;\mathrm{and}:\;\;\;\;\;\;\left\{\begin{array}{c} \min H_{S}{\left[\delta_{1}x_{1}+\delta_{2}x_{2}+\delta_{3}x_{3}+\delta_{4}x_{4}\right]}^{+}\\ \mathrm{subject}\;\mathrm{to}:\\ \;\;\;\;\;\;\;\;\;\;\;7.25x_{1}+8.5x_{2}+11.5x_{3}+7.75x_{4}\leq b\\ \;\;\;\;\;\;\;\;\;\;\;x_{1}+x_{2}+x_{3}+x_{4}=1\\ \;\;\;\;\;\;\;\;\;\;\;x_{i}\geq 0,\;i=1,2,3,4. \end{array}\right.$$

Similar to Example 15, the left model is a linear programming, and the right one is a nonlinear programming. By using MATLAB 2015a, the comparative results are obtained and shown in Table 4 and Figure 4. As we can see, no matter how much the upper limit is, the mean-entropy model keeps selecting Product 1, which has the potential to reach the lowest cost at three million dollars, and Product 2, which is most stable with the same maximum cost as Product 1. Besides, the higher the total cost that can be tolerated, the higher the proportion allocated to Product 2 is, since the expected total cost is closer to the expected cost of Product 2 and entropy dominates. By contrast, in the schemes obtained by the mean-semi-entropy optimization model, Product 1 accounts for more proportion, and Product 4 is included. As mentioned before, Product 1 is better than Product 2 from the perspective of cost; therefore, it is a better choice to invest more in Product 1. On the other hand, with the increase of the proportion distributed to Product 1, the total expected cost is decreased, and there are more chances to select the fluctuating Product 4. The semi-entropy optimization model can reach a much lower minimum total cost and therefore is recommended.

Together, we can see from the foregoing numerical examples that the results of the mean-entropy and mean-semi-entropy models are quite different, despite that they both rule out the alternatives with undesirable extreme values. Specifically, the mean-entropy model tends to construct concentrative schemes with stable alternatives, since entropy takes both sides of derivation as risks and excludes the alternatives with favorable extreme values mistakenly. Conversely, the mean-semi-entropy model prefers more distributive schemes, as semi-entropies measuring one-side risks retain the favorable extreme ones. As a consequence, the expectations of the mean-semi-entropy models are more desirable, i.e., the schemes derived from the mean-semi-entropy models have higher expected overall returns and lower expected total costs than those of the mean-entropy models.

### 5.4. Discussion

As shown above, entropy and semi-entropies can extensively serve as risk measures in FP for decision-making problems, whereas solving these models is incredibly complicated since the existing formulas are functions with uncertain measures. In comparison, based on the simplified calculation formulas proposed in this paper, the models are easily solved. Besides, when the objective and constraints in the optimization models are linear functions of the fuzzy numbers, the models can be defuzzified to the equivalent forms and solved by software more directly. Since lower and upper semi-entropies characterize bilateral uncertainty better, the mean-semi-entropy models turn out to be more reasonable with more distributed schemes, according to the results of Examples 15 and 16.

## 6. Conclusions

In the present work, we mainly concentrated on simplifying the calculations of entropy and semi-entropies in fuzzy environments and applying them in FP modeling as measures of weighing the uncertainty. All the fuzzy numbers δ in this paper were set as the regular LR type, which is most commonly used, and *f* was assumed to be a strictly monotone function of all the fuzzy numbers. The related work of the present work is summarized in Table 5, in which the contributions of this study (i.e., the concept of upper semi-entropy, the formulas of calculating the entropy and semi-entropies via the ICDs, the mean-entropy and mean-semi-entropy optimization models, as well as their equivalent models) are bold and colored.

To be specific, on the theoretical side, with the purpose of increasing the efficiency and expanding the application scope of these measures, some formulas for calculating the entropies and semi-entropies of a linear function with fuzzy numbers via the ICDs were proposed and verified. Compared with the formula via CF as Equation (Equation 15) (see Definition 6), calculating entropy by Equation (Equation 19) via ICD was simplified since the function *S* in the original definition was removed. Accordingly, the entropy of a linear function $\lambda_{1}\delta_{1}+\lambda_{2}\delta_{2}+\cdots +\lambda_{n}\delta_{n}$ can be calculated readily based on Equation (Equation 21). Besides, in comparison with the lower semi-entropy as Equation (Equation 22) proposed by Zhou et al. [39], which weighs downside risks, Equation (Equation 31) gives a simplified calculation formula similar to Equation (Equation 19) for entropy, and the formula for calculating the semi-entropy of a linear function can be expressed explicitly as Equation (Equation 31). Inspired by the lower semi-entropy, the concept of upper semi-entropy (Equation (Equation 25)) for measuring upside risk was put forward for the first time, and the calculating formula was derived as Equation (Equation 32).

With regard to the applications, contrary to Model (Equation 36) with a narrow application range and an incredibly complicated solving process induced by uncertain measures, two types of entropy optimization models by the calculation formulas above were formulated from a new perspective. First, Model (Equation 36) was generalized to all the return-oriented problems, and by the simplified formula Equation (Equation 19), it could be solved by a software package. Besides, owing to the good characterization of downside risks by lower semi-entropy, Model (Equation 37), which is supported by Equation (Equation 31), was proposed for comparison. When the objective and the constraints in the optimization models are linear functions of the fuzzy numbers, the models can be defuzzified to the equivalent forms as models (Equation 38) and (Equation 39) following from Equations (Equation 21) and (Equation 31) and solved by software directly. On the other hand, in terms of the cost-oriented problems, the models were formulated in a similar way. The mean-semi-entropy models were demonstrated to be more reasonable (refer to Examples 15 and 16).

To sum up, the primary results of this paper include the following aspects: (1) a formula for calculating the entropy of a regular LR fuzzy number via the CD and ICD was put forward, and it was verified that the entropy can be figured out in a simple way with the formula; (2) a formula for calculating the entropy of the widely-used linear function was proposed, which is quite general and can be extended to the cases where the fuzzy numbers are of different types; (3) the concepts of lower and upper semi-entropies, which have the advantage of measuring the one-side uncertainty, were introduced, and the corresponding formulas to calculate that of a linear function with regular LR fuzzy numbers were raised; (4) from the perspective of cost and return, respectively, two types of generalized entropy optimization models were proposed, and the equivalent formulations with regard to the special cases where the objective and constraints are linear functions of the fuzzy parameters were given, providing an effective method to modeling FP problems via entropy and semi-entropies.

## Figures and Tables

**Figure 1 entropy-21-00697-f001:**
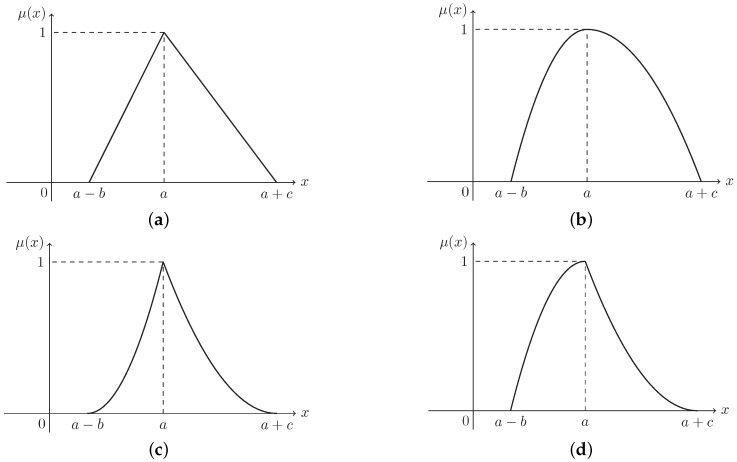
Membership functions (MFs) of the fuzzy numbers in Equations (Equation 2)–(Equation 5) in Examples 1–4. (**a**) Triangular $T{\left(a,b,c\right)}_{LR}$, (**b**) parabolic $P{\left(a,b,c\right)}_{LR}$, and (**c**) normal $N{\left(a,b,c\right)}_{LR}$, (**d**) mixture $M{\left(a,b,c\right)}_{LR}$.

**Figure 2 entropy-21-00697-f002:**
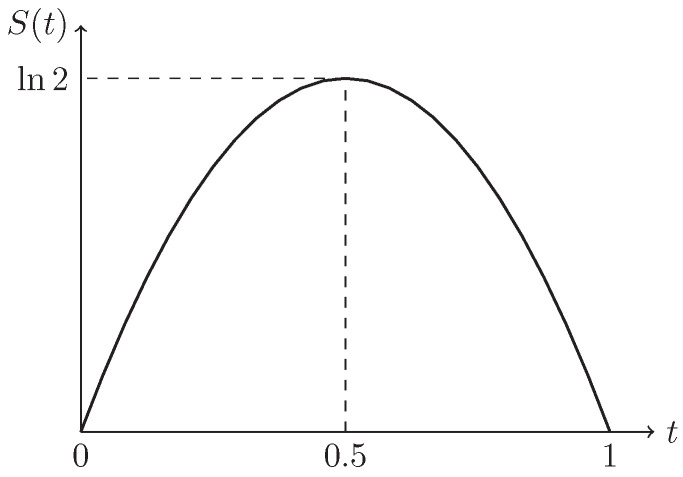
The function S(t) in Definition 6.

**Figure 3 entropy-21-00697-f003:**
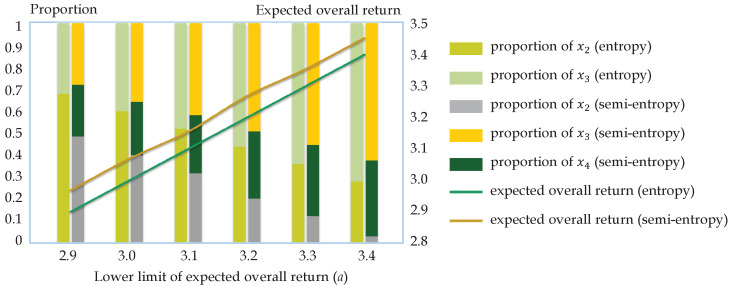
Optimal portfolios and expected overall returns of the portfolios derived by the mean-entropy and mean-semi-entropy optimization models under different lower limits *a* in Example 15. The bars in two (three) colors stand for the allocation schemes in the mean-entropy (mean-semi-entropy) model, and the line in brown (green) represents the expected return of each portfolio.

**Figure 4 entropy-21-00697-f004:**
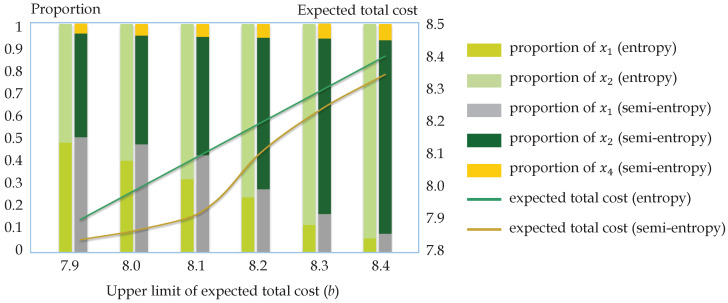
Optimal schemes and corresponding expected total costs obtained by the mean-entropy and mean-semi-entropy optimization models under different upper limits *b* in Example 16. The bars in two (three) colors stand for the allocation schemes in the mean-entropy (mean-semi-entropy) model, and the line in brown (green) represents the expected return of each scheme.

**Table 1 entropy-21-00697-t001:** Parameter settings, ranges of returns, expected values and entropies of $\delta_{1}$, $\delta_{2}$, $\delta_{3}$, $\delta_{4}$ in Example 15. The fuzzy numbers are all triangular fuzzy numbers in units of ten percent (e.g., the return of Stock 1 ranges from −60%–10%).

No.	Fuzzy Number	Range	Expected Value	Entropy
1	$\delta_{1}∼T{\left(-1,5,2\right)}_{LR}$	[−6,1]	−1.75	3.50
2	$\delta_{2}∼T{\left(2,1,3\right)}_{LR}$	[1,5]	2.50	2.00
3	$\delta_{3}∼T{\left(3,2,5\right)}_{LR}$	[1,8]	3.75	3.50
4	$\delta_{4}∼T{\left(3,6,6\right)}_{LR}$	[−3,9]	3.00	6.00

**Table 2 entropy-21-00697-t002:** Comparative results of the entropy optimization models in Example 15, including optimal portfolios, expected overall returns, and entropies (semi-entropies). With different expected returns *a*, the mean-entropy model keeps selecting Stocks 2 and 3 only, whereas the mean-semi-entropy model includes Stocks 2, 3, and 4 in the portfolios.

*a*	Mean-Entropy Model						Mean-Semi-Entropy Model					
	$x_{1}$	$x_{2}$	$x_{3}$	$x_{4}$	Mean	Entropy	$x_{1}$	$x_{2}$	$x_{3}$	$x_{4}$	Mean	Lower Semi-Entropy
2.90	0	0.68	0.32	0	2.90	2.48	0	0.48	0.28	0.24	2.97	1.54
3.00	0	0.60	0.40	0	3.00	2.60	0	0.40	0.36	0.24	3.07	1.61
3.10	0	0.52	0.48	0	3.10	2.72	0	0.32	0.42	0.26	3.15	1.69
3.20	0	0.44	0.56	0	3.20	2.84	0	0.20	0.49	0.31	3.27	1.83
3.30	0	0.36	0.64	0	3.30	2.96	0	0.12	0.55	0.32	3.35	1.91
3.40	0	0.28	0.72	0	3.40	3.08	0	0.03	0.62	0.34	3.45	2.01

**Table 3 entropy-21-00697-t003:** Parameter settings, expected values, and entropies of $\delta_{1}$, $\delta_{2}$, $\delta_{3}$, $\delta_{4}$ in Example 16. The fuzzy numbers are all triangular fuzzy numbers in units of millions of dollars (e.g., the R&D cost of Product 1 ranges from 3 million–10 million dollars).

No.	Fuzzy Number	Range	Expected Value	Entropy
1	$\delta_{1}∼T{\left(8,5,2\right)}_{LR}$	[3,10]	7.25	3.50
2	$\delta_{2}∼T{\left(9,3,1\right)}_{LR}$	[6,10]	8.50	2.00
3	$\delta_{3}∼T{\left(10,2,8\right)}_{LR}$	[8,18]	11.50	5.00
4	$\delta_{4}∼T{\left(8,6,5\right)}_{LR}$	[2,13]	7.75	5.50

**Table 4 entropy-21-00697-t004:** Comparative results of the entropy optimization models in Example 16, including optimal schemes, expected total costs, and entropies (semi-entropies). With different upper limits *b*, the mean-entropy model keeps selecting Products 1 and 2 only, whereas the mean-semi-entropy model selects Products 1, 2, and 4.

*b*	Mean-Entropy Model						Mean-Semi-Entropy Model					
	$x_{1}$	$x_{2}$	$x_{3}$	$x_{4}$	Mean	Entropy	$x_{1}$	$x_{2}$	$x_{3}$	$x_{4}$	Mean	Upper Semi-Entropy
7.90	0.48	0.52	0	0	7.90	2.72	0.50	0.46	0	0.04	7.84	1.26
8.00	0.40	0.60	0	0	8.00	2.60	0.47	0.48	0	0.05	7.87	1.25
8.10	0.32	0.68	0	0	8.10	2.48	0.42	0.52	0	0.06	7.93	1.24
8.20	0.24	0.76	0	0	8.20	2.36	0.28	0.66	0	0.06	8.11	1.14
8.30	0.16	0.84	0	0	8.30	2.24	0.17	0.77	0	0.06	8.24	1.08
8.40	0.08	0.92	0	0	8.40	2.12	0.08	0.85	0	0.07	8.34	1.03

**Table 5 entropy-21-00697-t005:** A summary of the calculation formulas and fuzzy programming models in this paper.

Means	Object	Methodology	Mechanism	Characteristic
Calculation formulas	Cr-based entropy	Equation (Equation 15)	Credibility function	Complex operation
		Equations (Equation 19) and (Equation 21)	Inverse credibility distribution	Simplified operation and favorable property of linear functions
	Semi-entropies	Equation (Equation 22)	Credibility function	Lower semi-entropy targeting to downside risks (complex operation)
		Equations (Equation 25), (Equation 31) and (Equation 32)	Inverse credibility distribution	Initializing concept of upper semi-entropy (simplified operation)
Fuzzy programming models	Return-oriented	Model (Equation 36)	Equation (Equation 15)	Model with uncertain measures (solved only by custom algorithms)
		Models (Equation 37)–(Equation 39)	Equations (Equation 19), (Equation 21), (Equation 31)	Mean-semi-entropy model and crisp equivalent forms (easily solved)
	Cost-oriented	Models (Equation 40)–(Equation 43)	Equations (Equation 19), (Equation 21), (Equation 32)	Novel perspective and equivalent models with simple solving process

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
