# Peer review of "Entropy and Semi-Entropies of LR Fuzzy Numbers’ Linear Function with Applications to Fuzzy Programming"

_entropy, 2019, doi:10.3390/e21070697_

Round 1
Reviewer 1 Report
The paper deals with the use of entropy and semi-entropies as risk measures and it provides a modeling approach for fuzzy programming.
The paper is very interesting and well done. I have just a comment: I suggest to the authors to extend and to highlight also the practical applications of the proposed method.
Author Response
We are extremely grateful to the two anonymous reviewers who have afforded us considerable assistance in enhancing both the quality of the findings and the clarity of their presentation. This manuscript has been revised substantially according to the issues raised by the editors and the referees. Please see the attachment for more details. Thank you. Jian

Reviewer 2 Report
This work provides a modeling method for fuzzy programming from the perspective of entropy.
And also it presents a framework for solving Fuzzy problems from the perspective of entropy and semi-entropies which are calculated in a simplified method. This manuscript is interesting and presents numerical examples to solve fuzzy problems.
My suggestion about the manuscript are:
Please check the journal format of the manuscript.
Please consider to add a Membership Function graph for each Fuzzy Set .
Another point, it is necessary reduce conclusions section and add the discuss in Entropy optimization models as a subsection.
Author Response

(The authors gave the same response as above.)
